# Heart Rate Variability Analysis on Electrocardiograms, Seismocardiograms and Gyrocardiograms of Healthy Volunteers and Patients with Valvular Heart Diseases

**DOI:** 10.3390/s23042152

**Published:** 2023-02-14

**Authors:** Szymon Sieciński, Ewaryst Janusz Tkacz, Paweł Stanisław Kostka

**Affiliations:** Department of Biosensors and Processing of Biomedical Signals, Faculty of Biomedical Engineering, Silesian University of Technology, F. D. Roosevelta 40, 41-800 Zabrze, Poland

**Keywords:** heart rate variability analysis, electrocardiography, seismocardiography, gyrocardiography, valvular heart diseases

## Abstract

Heart rate variability (HRV) is the physiological variation in the intervals between consecutive heartbeats that reflects the activity of the autonomic nervous system. This parameter is traditionally evaluated based on electrocardiograms (ECG signals). Seismocardiography (SCG) and/or gyrocardiography (GCG) are used to monitor cardiac mechanical activity; therefore, they may be used in HRV analysis and the evaluation of valvular heart diseases (VHDs) simultaneously. The purpose of this study was to compare the time domain, frequency domain and nonlinear HRV indices obtained from electrocardiograms, seismocardiograms (SCG signals) and gyrocardiograms (GCG signals) in healthy volunteers and patients with valvular heart diseases. An analysis of the time domain, frequency domain and nonlinear heart rate variability was conducted on electrocardiograms and gyrocardiograms registered from 29 healthy male volunteers and 30 patients with valvular heart diseases admitted to the Columbia University Medical Center (New York City, NY, USA). The results of the HRV analysis show a strong linear correlation with the HRV indices calculated from the ECG, SCG and GCG signals and prove the feasibility and reliability of HRV analysis despite the influence of VHDs on the SCG and GCG waveforms.

## 1. Introduction

Cardiovascular diseases remain the most common cause of death in the world and constitute a significant concern for public health due to the economic burden (expected to reach 47 trillion USD by 2030) and the overload on medical personnel (17.9 million deaths worldwide in 2016) despite progress in prevention, diagnosis and therapy [1,2,3,4,5]. Due to its growing prevalence and significant impact on quality of life [6], we consider valvular heart disease (VHD) in this study.

Valvular heart disease is any cardiovascular disease that affects any heart valve (the aortic valve, mitral valve, pulmonic valve and tricupsid valve) [7,8]. The main causes of VHDs are rheumatic heart disease and ageing [6,8,9,10]. The most prevalent VHD is aortic stenosis (AS), which is the third most common cardiovascular disease after hypertension and coronary artery disease, and is usually caused either by degenerative calcification of the aortic valve or progressive stenosis of a congenital bicuspid valve [8].

VHDs are usually diagnosed by echocardiography, computed tomography or magnetic resonance imaging [9], which are not feasible in outpatient monitoring [11]. This problem has been addressed by applying various methods [12,13], including exercise electrocardiography (ECG) [14], seismocardiography (SCG) or gyrocardiography (GCG), which register the current mechanical function of the heart with an inertial measurement unit (IMU) placed on the chest wall [11,15,16].

Seismocardiography (SCG) and gyrocardiography (GCG) are two complementary techniques [15,17,18]; seismocardiography is a technique for registering low-frequency precordial acceleration, invented by Bozhenko in 1961 [19,20], whereas gyrocardiography registers the rotational component of cardiac vibrations and was invented by Meriheinä et al. in 2015 [15,16,18,21,22].

Seismocardiographic (SCG) and gyrocardiographic (GCG) signals are non-stationary signals with distinct quasiperiodic features known as waves, e.g., the mitral valve opening wave (MO), the mitral valve closure wave (MC), the isovolumetric contraction wave, the rapid ejection wave, the aortic valve opening wave (AO), the aortic valve closure (AC) wave and the cardiac filling wave [18,22,23,24,25,26,27,28]. Figure 1 presents an annotation of concurrent ECG, SCG and GCG signals in a healthy subject.

SCG and GCG have found applications in the diagnosis of several cardiovascular diseases, such as aortic stenosis [30,31,32], aortic valve disease (AVD) [33], coronary artery disease [34], myocardial infarction [35,36], atrial fibrillation [37,38,39,40], the effects of cardiac resynchronization therapy [41] and heart failure [35,42]. This has usually involved the use of computational intelligence techniques, such as artificial neural networks (including deep and convolutional neural networks), random forests, extreme gradient boosting and support vector machines [28,30,32,35,36,37,38,39,40].

One of the most prominent applications of seismocardiography and gyrocardiography is heart rate variability (HRV) analysis [16,18,22,31,43,44,45,46,47,48,49,50,51,52,53,54,55]. Heart rate variability is defined as the physiological variation in the intervals between consecutive heartbeats (inter-beat interval) and reflects the activity of the autonomic nervous system [56,57,58].

HRV analysis has traditionally been performed on interbeat intervals obtained from electrocardiograms (ECG signals) [22,43,45,50,51,55,56,59]. The first attempt of HRV analysis based on cardiac mechanical signals (mechanocardiograms) was performed by Friedrich et al. in 2010 [60] on ballistocardiograms. In 2012, Ramos-Castro et al. performed the first HRV analysis on seismocardiograms [43], and the first HRV analysis on gyrocardiograms was performed by Lahdenoja et al. [61] in 2016. The validity of HRV indices obtained from the SCG signal was first demonstrated by Laurin et al. [62] in 2013 and then in later studies [22,45,46,51,52,53,54,55].

The advantages of using seismocardiography and/or gyrocardiography over electrocardiography for cardiac diagnosis are the simpler measurement setup (using only one sensor) and the availability of information on cardiac intervals, contractility and the state of heart valves at the same time [11,16,22,27,43,44,45,51,53,56,59,63]. However, the limitations of SCG and GCG include the inter-subject variability of signal morphology that can be significantly affected by cardiac diseases or sensor placement and susceptibility to motion artifacts [11,16,23,27,55,64].

The purpose of this study is to evaluate the differences between the time domain, frequency domain and nonlinear HRV indices derived from electrocardiograms, seismocardiograms and gyrocardiograms in healthy volunteers and patients with valvular heart diseases. This study is an extended version of [54] and is based on two publicly available datasets obtained from healthy people and patients with VHDs.

## 2. Materials and Methods

### 2.1. Datasets

This study was carried out on two publicly available datasets with concurrent electrocardiograms, seismocardiograms and gyrocardiograms. The first dataset if “Mechanocardiograms with ECG reference” published by M. Kaisti et al. [65,66] containing signals acquired from twenty-nine healthy volunteers and the second contains thirty signals derived from “An Open-access Database for the Evaluation of Cardio-mechanical Signals from Patients with Valvular Heart Diseases” published by C. Yang et al. in [11,67].

The first dataset consists of 29 recordings of concurrent ECG, SCG and GCG signals acquired from 29 healthy male volunteers that were registered with sensors attached to the chest wall over the sternum with a double-sided tape and with a sampling frequency of 800 Hz. The subjects were lying either in the supine position or on their left or right side [61,65,66].

Electrocardiograms were acquired with an ADS1293 (Texas Instruments, Dallas, TX, USA), seismocardiograms were recorded with a triaxial capacitive digital accelerometer (MMA8451Q from Freescale Semiconductor, Austin, TX, USA) and gyrocardiograms were acquired using a 3-axial MAX21000 gyroscope (Maxim Integrated, San Jose, CA, USA) [15,65].

The rotation and translation axes in seismocardiography and gyrocardiography were defined for both datasets as follows: the x axis was oriented laterally from left to right, the y axis was oriented from head to foot and the z axis was oriented from back to chest [65].

The second dataset consists of 100 simultaneous recordings of raw ECG, SCG and GCG signals with annotated heartbeats acquired from 100 patients with valvular diseases admitted to two different clinical sites: 30 patients were admitted to Columbia University Medical Center (New York City, NY, USA) and 70 patients were admitted to the First Affiliated Hospital of Nanjing Medical University (Nanjing, Jiangsu Province, People’s Republic of China). ECG, SCG and GCG signals were recorded before any treatment in both populations of patients [11].

To balance the number of healthy volunteers and patients with valvular heart disease in this comparison study, we took only 30 patients (14 female and 16 male subjects) who were admitted to the Columbia University Medical Center (New York City, NY, USA). A total of thirty patients had aortic stenosis, nine patients had tricupsid valve regurgitation (TR), five had mitral valve stenosis (MS), four had mitral valve regurgitation (MR) and no patients had aortic valve regurgitation. During registration, each subject was asked to be awake and stay in the supine position, breathing normally.

The ECG, SCG and GCG signals were recorded with Shimmer 3 ECG module (Shimmer Sensing, Dublin, Ireland) with a sampling frequency of 256 Hz (recordings UP-01 to UP-21) and 512 Hz (recordings UP-22 to UP-30) [11,67]. The shimmer 3 device contains a 3-axial inertial measurement unit that contains an accelerometer, a gyroscope and a magnetometer (ICM-20948 from TDK InvenSense, San Jose, CA, USA) and a separate low-noise 3-axial Kionix KXTC9-2050 accelerometer (Kionix, Inc., Ithaca, NY, USA) [68]. Before the measurements, each subject gave informed consent by signing a consent form. All metadata were deidentified before publication [11,67].

The basic characteristics of both datasets are shown in Table 1. Figure 2 and Figure 3 present a 15-second and 16-second fragment of raw ECG, SCG and GCG signals in subject 10 from the first dataset and UP-13 of the second dataset, respectively. More details are revealed in Appendix B.

### 2.2. Signal Processing

Signal processing started with importing the data into MATLAB R2022b (MathWorks, Inc., Natick, MA, USA); data from [67] were directly loaded into the MATLAB workspace, while data from [66] required importing each line of the text files containing signal samples for subjects 1–8 and discarding the first sample (an artifact) for subjects 9–29. In both datasets, each file represented one subject [66,67].

Heartbeat detection in SCG signals and GCG signals for both datasets was based on the approach presented in [11,22,45,53,54,69], which consists of the following steps: The first step was to apply the Pan–Tompkins algorithm (introduced in [70]) to ECG signals. The next step was to find local maxima in the SCG and GCG signals within 100 ms of the closest R waves in the ECG signals based on the observations published in [45,71], and the final step was to calculate the intervals between each consecutive heartbeat in the ECG, SCG and GCG signals [11,69]. An example of a tachogram for a healthy subject is shown in Figure 4, and Figure 5 presents a tachogram of a patient with VHD.

Local maxima that occur within 100 ms of the R wave in concurrent ECG signals are associated with the aortic valve opening waves that are single sharp peaks on the z-axis of the SCG signal and the y-axis of the GCG signal [15,18,20,45,71]. Taking into account only the z-axis of the SCG signal and the y-axis of the GCG signal for analyses was based on the higher signal-to-noise ratio compared to the other axes [15,22,23,65].

### 2.3. HRV Analysis

HRV analysis was carried out according to the recommendations published in [56,59] and consisted of the following time and frequency domain indices: mean interbeat interval (AVNN); standard deviation of the interbeat interval (SDNN); root mean square of the differences of successive interbeat intervals (RMSSD); the ratio of successive differences greater than 50 ms in all interbeat intervals (RMSSD); the power of the HRV signal in the very low-frequency band (VLF), in the low-frequency band (LF) and in the high-frequency band (HF); and the LF/HF ratio (LF/HF).

The frequency bands of the HRV spectrum were defined as follows: the very low-frequency band was defined as 0.0033–0.04 Hz, the low frequency band was defined as 0.04–0.15 Hz and the high-frequency band was defined as 0.15–0.4 Hz [56,72]. The analyses were performed with the CardioNet Cardiovascular Signal Toolbox and MATLAB R2022b. HRV indices in the frequency domain were based on spectral power estimates calculated as 1024-sample Lomb periodograms [72,73]. The Lomb periodogram for frequency ω is expressed as:(1)$$P_{x}\left(\omega \right)=\frac{1}{2}\left(\frac{{\left[\sum_{j}X_{j}\cos\omega \left(t_{j}-\tau \right)\right]}^{2}}{\sum_{j}\cos^{2}\omega \left(t_{j}-\tau \right)}+\frac{{\left[\sum_{j}X_{j}\sin\omega \left(t_{j}-\tau \right)\right]}^{2}}{\sum_{j}\sin^{2}\omega \left(t_{j}-\tau \right)}\right)$$
where τ is the time delay defined in Equation (Equation 2), $X_{j}$ is the value of the *j*-th sample and $t_{j}$ is the time of the *j*-th sample [74,75].
(2)$$\tan2\omega \tau =\frac{\sum_{j}\sin2\omega t_{j}}{\sum_{j}\cos2\omega t_{j}}.$$

The non-linear analysis of heart rate variability was based on three indices derived from the geometrical features of Poincaré maps: SD_1_, SD_2_ and SD_1_/SD_2_.

SD_1_ is a measure of the short-term heart rate variability that is defined as the width of an ellipse fitted to scatter points of a Poincaré map and may be expressed as the standard deviation of the distances from the identity line (y=x axis) of the Poincaré plot [76,77]:(3)$${\mathrm{SD}}_{1}=stddev\left(\frac{\left|NN_{i+1}-NN_{i}\right|}{\sqrt{2}}\right)$$
where $NN_{i}$ is the *i*-th inter-beat interval series for i=1,2,…N−1, $NN_{i+1}$ is the next inter-beat interval and stddev() denotes the standard deviation (SD) [78,79,80,81].

SD_2_ is the length of an ellipse fitted to the scatter points of a Poincaré map that reflects the long-term heart rate variability and is calculated as the standard deviation of the distance of points from the $y=-x+2\bar{NN}$ axis:(4)$${\mathrm{SD}}_{2}=stddev\left(\left|\frac{NN_{i+1}-NN_{i}}{\sqrt{2}}-2\bar{NN}\right|\right).$$

SD_1_/SD_2_ is calculated as the ratio between SD_1_ and SD_2_ and reflects the unpredictability of the heart rate [82].

## 3. Results

The results of HRV analyses on electrocardiograms, seismocardiograms and gyrocardiograms obtained from healthy volunteers and patients with VHDs were expressed as the mean and standard deviation (SD) values and are shown in Table 2, Table 3 and Table 4, respectively. The HRV indices calculated for patients with VHD were derived from [53], except for SD_1_, SD_2_ and SD_1_/SD_2_.

The mean and standard deviation values of most HRV indices for patients with VHD are significantly different from those of healthy volunteers, except for AVNN and VLF. These differences were further evaluated by applying the Student’s *t*-test for the significance level of 0.05. The results of the *t*-test are shown in Table 5.

The differences between the HRV indices in healthy volunteers and in patients with VHD shown in Table 5 are statistically significant for all analyzed HRV indices except for AVNN, SDNN, pNN50 (ECG), VLF (GCG), LF and SD_2_. This proves a significant influence of ventricular heart diseases on the results of heart rate variability in time domain, frequency domain and nonlinear analyses. These results confirm the findings related to Table 3.

The findings reported in [17,22,45,51] for HRV indices obtained from ECG, SCG and GCG signals in healthy volunteers and patients with VHDs were verified with the Pearson’s linear correlation that were expressed as Pearson’s linear correlation coefficient (ρ) for healthy volunteers and VHD patients. A linear correlation coefficient larger than 0.7 was considered as a strong linear correlation between two given datasets [83].

The results presented in Table 6 indicate a strong linear correlation for the *p*-value under 0.001, except for SD_1_/SD_2_ between ECG and SCG signals from VHD patients. The correlation between the analyzed HRV indices obtained from ECG and GCG signals (as shown in Table 7) is weaker than between the ECG and SCG signals, but remains strong for all HRV indices except for VLF, SD_2_ and SD_1_/SD_2_. The strongest correlation is observed for HF, pNN50, RMSSD and SD_1_, and the weakest correlation is observed for VLF (−0.0663).

## 4. Discussion

We have performed HRV analysis on electrocardiograms, seismocardiograms and gyrocardiograms from healthy volunteers and patients with VHD based on publicly available datasets.

The mean and standard deviation values of HRV indices obtained from healthy subjects are similar to those reported by Siecinski et al. in [22,50,52,53], except for the LF/HF and frequency domain indices in [22,52], and also similar to the results reported by Ramos-Castro et al. in [43] and Tadi et al. in [45]. The discrepancies in the mean HRV indices are within the standard deviation for each signal (ECG, SCG and GCG) and may be related to inter-subject variations.

The mean and standard deviation values of most HRV indices for patients with VHD are significantly different from those of healthy volunteers, except for AVNN, LF and SD_2_. This observation was confirmed by a Student’s t-test. Despite the fact that RMSSD and SD_1_ should be identical [76], there were significant differences between these indices in each case.

We have shown the significant influence of valvular heart disease on HRV indices, except for AVNN, SDNN (ECG and SCG), pNN50 (ECG), VLF (GCG), LF, HF (ECG) and SD_2_, which was in line with [84,85]. The similarities between the results of the HRV analysis in patients with VHD in our study and those reported in [84] prove that the HRV indices obtained from seismocardiograms are valid both for healthy subjects, patients with aortic stenosis and also for other VHD patients as long as heartbeats were reliably detected [45,53,54,62].

Despite the significant influence of VHDs on HRV indices, the correlation between the HRV indices obtained from ECG and SCG signals is strong, except for SD_2_. Obtained values of ρ are similar to those reported by Siecinski et al. in [50] and Charlier et al. in [86]. However, the correlations of analyzed pairs of signals are weaker for VHD patients than for healthy volunteers, especially the correlations of HRV indices from ECG and GCG signals [22,43,45,52].

Such results are influenced by age (the population of healthy volunteers is significantly younger than those of patients with VHD), comorbidities, signal quality and the use of different accelerometers and gyroscopes operating with different sampling frequency and accuracy, according to the available datasheets [68,87,88].

The differences between RMSSD and SD_1_ values did not result in significantly different ρ values. This indicates the lower accuracy of automatic heartbeat detection in SCG and GCG signals of patients with VHDs that was affected by morphological changes caused by VHDs and/or ageing [53,54].

The limitations of the study include the use of only one type of heartbeat detector for seismocardiograms and gyrocardiograms that depends on a concurrent electrocardiogram, the influence of specific cardiovascular conditions or medication on the calculated HRV indices was not considered and the morphological changes in SCG and GCG signals due to valvular heart disease was not evaluated.

In future studies, we will consider evaluating the influence of various cardiovascular conditions on HRV indices derived from ECG, SCG and GCG signals; indices derived from larger and more diverse groups, including the analysis of SCG and GCG signal morphology; and indices derived from other detectors of SCG and GCG signals. In this study, we proved that a heart rate variability analysis based on cardiac mechanical signals [30,31,89] may be useful for a more cost-effective and convenient diagnosis and monitoring of patients with cardiovascular disease.

## 5. Conclusions

The results of the heart rate variability analysis based on mechanocardiograms (SCG and GCG signals) in both a healthy population and patients with VHD remain valid as long as heartbeats are correctly detected. Valvular heart disease significantly affects RMSSD, pNN50 (only SCG and GCG signals), VLF (only ECG and SCG signals), HF (only SCG and GCG signals), LF/HF, SD_1_ and SD_1_/SD_2_. The linear correlation between the HRV indices obtained from the ECG and the mechanocardiograms is strong both in healthy volunteers and in patients with VHD, except for SD_2_.

Future studies should include an evaluation of other cardiovascular conditions, larger and more diverse groups and other heartbeat detection methods for mechanocardiograms, and an analysis of the morphology of cardiac mechanical signals. We showed that mechanocardiogram-based heart rate variability analyses can be used in the diagnosis and monitoring of cardiovascular disease, which could be more cost-effective and convenient for patients.

## Figures and Tables

**Figure 1 sensors-23-02152-f001:**
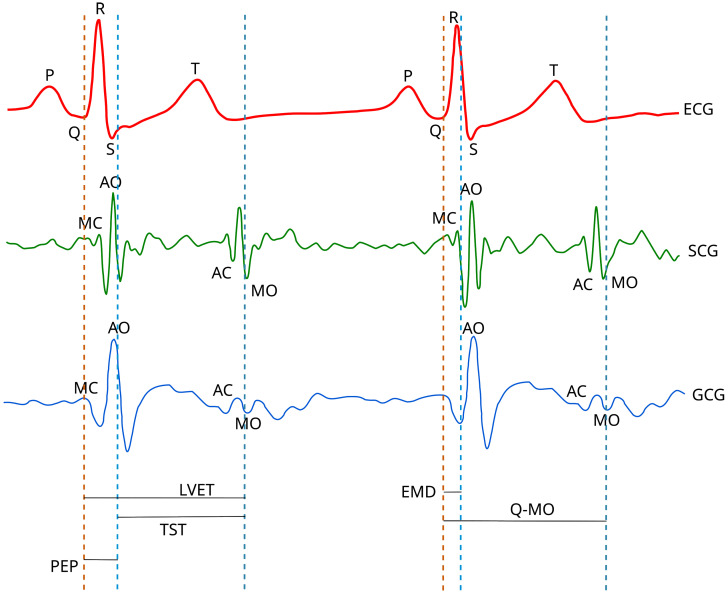
The annotation of ECG, SCG and GCG waveforms in healthy subjects. Based on the diagrams published in [15,18,26,29] under the CC-BY 4.0 license.

**Figure 2 sensors-23-02152-f002:**
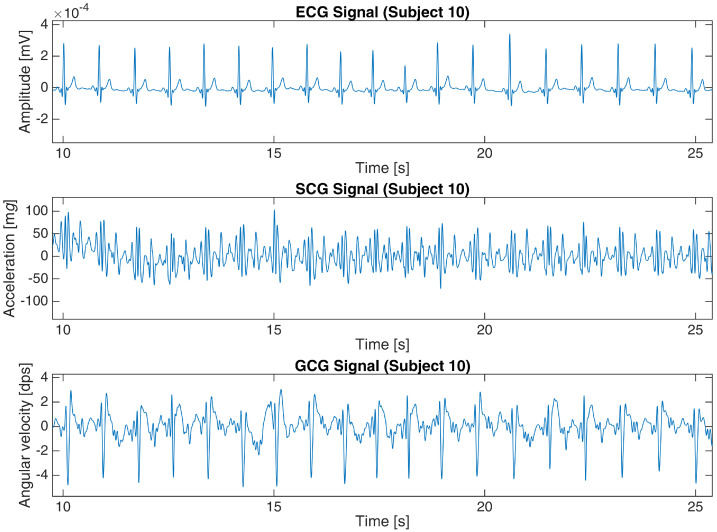
ECG, SCG and GCG signals from subject 10 in the first dataset (25-s fragment).

**Figure 3 sensors-23-02152-f003:**
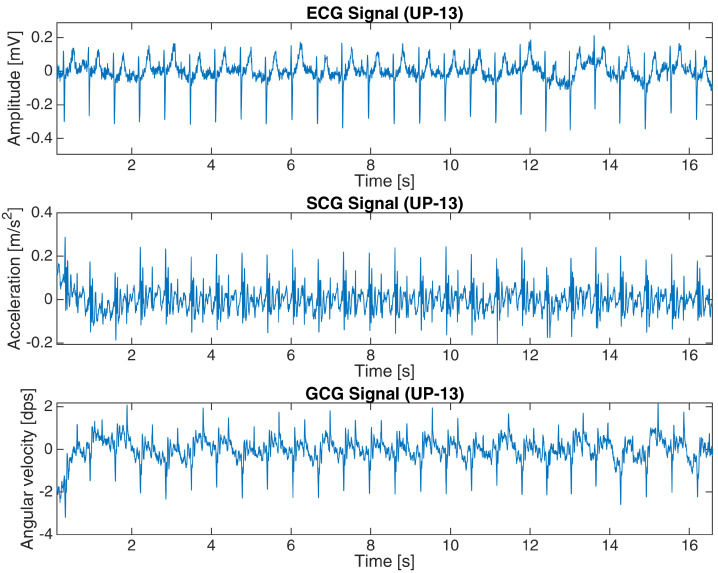
ECG, SCG and GCG signals from subject UP-13 in the second dataset (16-s fragment).

**Figure 4 sensors-23-02152-f004:**
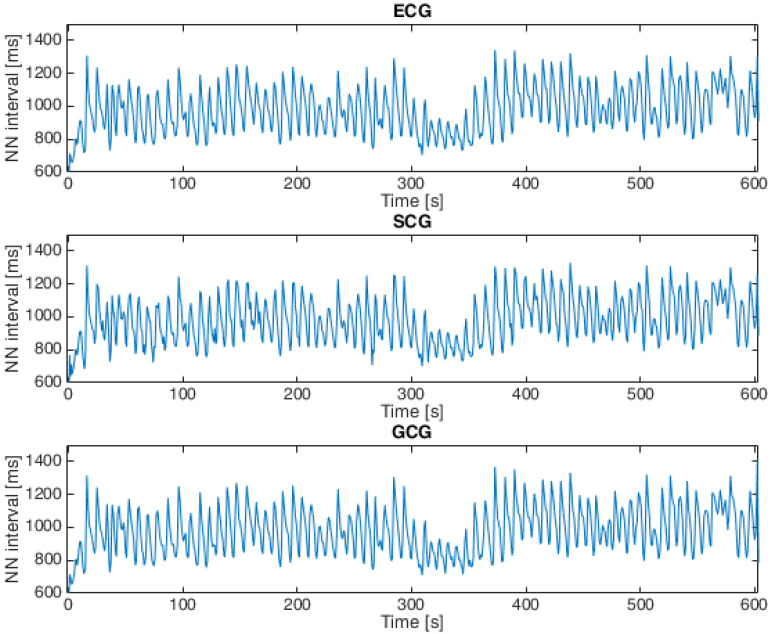
Tachogram derived from ECG, SCG and GCG signals taken from subject 9 in the first dataset (15-second fragment).

**Figure 5 sensors-23-02152-f005:**
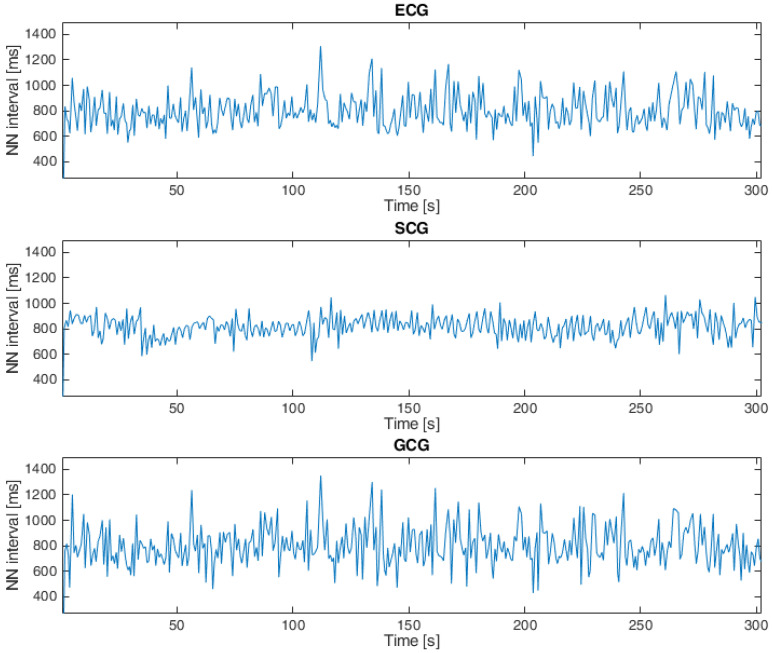
Tachogram ECG, SCG and GCG signals taken from subject UP-09 in the second dataset.

**Table 1 sensors-23-02152-t001:** Basic characteristics of the datasets. Aggregated values are expressed as the range (min–max) and mean ± SD.

Dataset	Number of	Age	Height	Weight	BMI	Recording
	Subjects	(years)	(cm)	(kg)	(kg/m^2^)	Time (min)
Healthy	29 male	23–41	170–190	60–98	18–30	253
population		29 ± 5	179 ± 5	76 ± 11	24 ± 3	
VHD	14 female	68–97	140–183	44–118	19–40	174.8
patients	16 male	83 ± 8	163 ± 12	74 ± 17	28 ± 6	
	(30 in total)					

**Table 2 sensors-23-02152-t002:** HRV indices derived from ECG signals from both datasets.

HRV Index	Healthy		VHDs	
	Mean	SD	Mean	SD
AVNN (ms)	952.2551	112.1082	881.7178	155.9992
SDNN (ms)	93.7994	32.0249	94.7063	47.4722
RMSSD (ms)	84.7391	36.1640	121.6602	74.7506
pNN50	0.3092	0.1924	0.3152	0.3178
VLF (ms^2^)	2108.3429	1555.0081	960.4883	828.3130
LF (ms^2^)	2947.2316	2468.9979	2190.3676	2270.7844
HF (ms^2^)	3493.6581	2550.7361	5687.0552	5676.3914
LF/HF	0.9345	0.5333	0.4307	0.1792
SD_1_ (ms)	59.9739	26.1703	86.1176	52.9350
SD_2_ (ms)	117.6626	39.0786	101.2172	44.6326
SD_1_/SD_2_	0.5026	0.1258	0.8095	0.2275

**Table 3 sensors-23-02152-t003:** HRV indices derived from SCG signals from both datasets.

HRV Index	Healthy		VHDs	
	Mean	SD	Mean	SD
AVNN (ms)	952.2583	112.1185	881.5849	156.4511
SDNN (ms)	96.7361	31.9037	113.0716	40.8948
RMSSD (ms)	92.8507	37.1027	160.9644	63.2959
pNN50	0.3590	0.1794	0.5499	0.2345
VLF (ms^2^)	2108.0188	1559.9375	1009.8038	849.8141
LF (ms^2^)	2967.0571	2477.1760	2413.8259	2320.6393
HF (ms^2^)	3898.4718	2926.5900	7275.5874	5670.2440
LF/HF	0.8986	0.5179	0.3177	0.1617
SD_1_ (ms)	65.7216	28.4287	113.9518	44.8231
SD_2_ (ms)	119.3105	39.8927	110.7745	40.7536
SD_1_/SD_2_	0.5437	0.1265	1.0515	0.3080

**Table 4 sensors-23-02152-t004:** HRV indices derived from GCG signals from both datasets.

HRV Index	Healthy		VHDs	
	Mean	SD	Mean	SD
AVNN (ms)	952.2358	113.3623	929.9744	222.8833
SDNN (ms)	86.6979	31.6025	133.0636	66.8667
RMSSD (ms)	83.6785	36.3714	183.8181	79.3316
pNN50	0.3712	0.1717	0.5551	0.2799
VLF (ms^2^)	2119.8767	1568.1258	3880.6816	13,313.9379
LF (ms^2^)	2978.3266	2484.0123	3251.1583	3594.0590
HF (ms^2^)	3663.0536	2657.3141	9481.6615	7681.8666
LF/HF	0.8997	0.5328	0.3217	0.1611
SD_1_ (ms)	63.7232	26.2834	130.1497	56.1906
SD_2_ (ms)	118.6764	39.0182	130.1935	80.8863
SD_1_/SD_2_	0.5347	0.1369	1.0302	0.2766

**Table 5 sensors-23-02152-t005:** Results of Student’s *t*-tests between healthy subjects and VHD subjects.

HRV Index	ECG		SCG		GCG	
	h *	*p*-Value	h *	*p*-Value	h *	*p*-Value
AVNN	0	0.0516	0	0.0516	0	0.6316
SDNN	0	0.9320	0	0.0748	1	0.0085
RMSSD	1	0.0201	1	<0.0001	1	<0.0001
pNN50	0	0.8544	1	<0.0001	1	<0.0001
VLF	1	<0.0001	1	0.0012	0	0.4823
LF	0	0.5215	0	0.3742	0	0.7365
HF	0	0.0621	1	0.0028	1	<0.0001
LF/HF	1	<0.0001	1	<0.0001	1	<0.0001
SD_1_	1	0.0201	1	<0.0001	1	<0.0001
SD_2_	0	0.4502	0	0.6924	0	0.3863
SD_1_/SD_2_	1	<0.0001	1	<0.0001	1	<0.0001

* h = 0 means no significant difference.

**Table 6 sensors-23-02152-t006:** Pearson’s linear correlation coefficient of HRV indices obtained from ECG and SCG signals.

HRV Index	ρ (Healthy Subjects)	ρ (VHD Subjects)
AVNN	1.0000	0.9999
SDNN	0.9942	0.8767
RMSSD	0.9754	0.8164
pNN50	0.6402	0.7026
VLF	0.9999	0.8867
LF	0.9996	0.9390
HF	0.9868	0.9493
LF/HF	0.9916	0.7296
SD_1_	0.9754	0.8164
SD_2_	0.9980	0.9364
SD_1_/SD_2_	0.9375	0.4116

**Table 7 sensors-23-02152-t007:** Pearson’s linear correlation coefficient of HRV indices obtained from ECG and GCG signals.

HRV Index	ρ (Healthy Subjects)	ρ (VHD Subjects)
AVNN	1.0000	0.5602
SDNN	0.9942	0.4830
RMSSD	0.9754	0.6134
pNN50	0.6402	0.6497
VLF	0.9999	−0.0663
LF	0.9996	0.5105
HF	0.9842	0.6818
LF/HF	0.9906	0.6531
SD_1_	0.9976	0.6132
SD_2_	0.9998	0.3829
SD_1_/SD_2_	0.9841	0.3684

## Data Availability

The research was based on publicly available data published in [66,67]. Appendix A (source code and full results) are available online. However, the data presented in this study are available on request from the corresponding author.

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
