# Peer review of "Heart Rate Variability Analysis on Electrocardiograms, Seismocardiograms and Gyrocardiograms of Healthy Volunteers and Patients with Valvular Heart Diseases"

_sensors, 2023, doi:10.3390/s23042152_

Round 1

Reviewer 1 Report

This paper is interesting and it is a logical follower of the earlier papers from the same authors.

I would like to propose the following corrections/improvements to the manuscript:

-It appears that in Figure 4 the blue curves for SCG and GCG are identical. Please check if you accidentally used the same tachogram in both SCG and GCG.
-In Figures 4 and 5 the rightmost part of the plots should be cropped for example to approximately x=610 seconds and and x=320 seconds, respectively in order to remove the blank space from the right. In addition in Figures 4 and 5, please consider the possibility to limit the y-axis into intervals of 500-1500 and 500-1000, respectively to remove the blank space from the bottom.
-In Table 2 regarding the valvular dataset (or in text) the authors should (in addition to individual statistics) report the mean and STD of the overall demographics parameters (age, height, weight, BMI).
-I would request the authors to add two new tables to the manuscipt. First, for the healthy volunteers dataset (containing a total of 29 rows plus title row) and second for the Columbia University Medical Center valvular heart diseases dataset (a total of 30 rows plus title row). As rows there would be individual patient ID's (subject 1-29 in first table and subject numbers UP-01-UP-30 in the second table) on the lestmost column. In both of these tables there would be columns for AVNN and SDNN and RMSSD value, extracted from ECG, SCG and GCG. This presentation would allow a more comprehensive analysis and comparison of the HRV metrics for each individual.
-This paper (in part) uses the same dataset as in the same author's older papers. Thus, the authors should briefly mention any possible reasons if some HRV metrics within the same dataset differ from their earlier papers (e.g. caused by different version of Matlab or HRV library or peak detection algorithm).
-In Table 8 correlation between ECG and GCG in valvular diseases dataset was relatively low (in comparison with healthy individuals dataset). The authors should investigate this issue a bit more if a reason for this can be found. In my personal view the most probable reason for this is different gyroscope sensor that was used in the valvular diseases dataset (most probable it is not of exceptionally high quality). In principle there
could also be some other reason such as unintended usage of different axis in different datasets or then there might be some additional delay between ECG and GCG in the different datasets. The authors could try to assess the differences between these two different gyroscope models (Shimmer 3's gyroscope in VHD dataset and Maxim integrated MAX21000 sensor in healthy dataset) from their datasheets and comment whether this is the case with one or two sentences in the discussion.
-It is quite well known that usually (in average) the HRV of a person decreases when age increases and HRV may decrease when patient has cardiovascular health problems. Also a lower heart rate (HR) may
imply more efficient heart function and better cardiovascular health (at least in healthy individuals) with the exception is that low heart rate (HR) may also relate to use of certain medications (e.g. beta blockers) in diseased individuals. Thus in the discussion section the authors should discuss the changes observed in ECG in HR and HRV between the heathy volunteers dataset and the valvular diseases dataset in the light of these four aspects. For instance, the healthy volunteers are much younger than valvular disease group and this may reflect to HR and HRV and on the other hand patients in valvular disease dataset may have medications which might affect to HR/HRV and also different gender distribution in the two datasets may affect to average HR/HRVs etc. I would recommend to add a few additional sentences about these issues to the discussion section (potentially supplemented with a couple of new citations).
-As a last section of the manuscript (section 5) I would recommend to add conclusions briefly summarising the main findings of this paper.

Typo corrections:
-In Figure 4 Tachogramm word there is a typo (extra "m").
-Line 192 (Discussion): ...SDNN, (ECG and SCG) pNN50 (ECG), VLF (GCG)... There is probably missing comma between some of the words.
-In section 2.3 lines 139-140 there is a text: "The very low-frequency band of the HRV spectrum was defined as 0.0033–0.04 Hz, low frequency band was defined as 0.04–0.15 Hz, and the high-frequency band was defined as
0.15–0.4 Hz [54,68].". This sentence should be revised for clarity.
-The paper should be carefully read through once more in order to correct any other possible typos.

Author Response

This paper is interesting and it is a logical follower of the earlier papers from the same authors.

Answer: We confirm maintaining the logical order from earlier papers

I would like to propose the following corrections/improvements to the manuscript:

-It appears that in Figure 4 the blue curves for SCG and GCG are identical. Please check if you accidentally used the same tachogram in both SCG and GCG.

Answer: Thank you for spotting that mistake. We have replaced the old version with the corrected version.

-In Figures 4 and 5 the rightmost part of the plots should be cropped for example to approximately x=610 seconds and and x=320 seconds, respectively in order to remove the blank space from the right. In addition in Figures 4 and 5, please consider the possibility to limit the y-axis into intervals of 500-1500 and 500-1000, respectively to remove the blank space from the bottom.

Answer: The figures have been corrected.

-In Table 2 regarding the valvular dataset (or in text) the authors should (in addition to individual statistics) report the mean and STD of the overall demographics parameters (age, height, weight, BMI).

Answer: We have added the overall demographic parameters of the VHD patient data set.

-I would request the authors to add two new tables to the manuscipt. First, for the healthy volunteers dataset (containing a total of 29 rows plus title row) and second for the Columbia University Medical Center valvular heart diseases dataset (a total of 30 rows plus title row). As rows there would be individual patient ID's (subject 1-29 in first table and subject numbers UP-01-UP-30 in the second table) on the lestmost column. In both of these tables there would be columns for AVNN and SDNN and RMSSD value, extracted from ECG, SCG and GCG. This presentation would allow a more comprehensive analysis and comparison of the HRV metrics for each individual.

Answer: We have not added two new tables with complete HRV results to the main body of the article due to the size. We provide these tables as a supplementary material. 

-This paper (in part) uses the same dataset as in the same author's older papers. Thus, the authors should briefly mention any possible reasons if some HRV metrics within the same dataset differ from their earlier papers (e.g. caused by different version of Matlab or HRV library or peak detection algorithm).

Answer: HRV indices obtained from healthy population slightly differ due to the different preprocessing algorithm. 

-In Table 8 correlation between ECG and GCG in valvular diseases dataset was relatively low (in comparison with healthy individuals dataset). The authors should investigate this issue a bit more if a reason for this can be found. In my personal view the most probable reason for this is different gyroscope sensor that was used in the valvular diseases dataset (most probable it is not of exceptionally high quality). In principle there
could also be some other reason such as unintended usage of different axis in different datasets or then there might be some additional delay between ECG and GCG in the different datasets. The authors could try to assess the differences between these two different gyroscope models (Shimmer 3's gyroscope in VHD dataset and Maxim integrated MAX21000 sensor in healthy dataset) from their datasheets and comment whether this is the case with one or two sentences in the discussion.

Answer: We used the same axis for SCG (z-axis) and GCG (y-axis) in both data sets. 

-It is quite well known that usually (in average) the HRV of a person decreases when age increases and HRV may decrease when patient has cardiovascular health problems. Also a lower heart rate (HR) may
imply more efficient heart function and better cardiovascular health (at least in healthy individuals) with the exception is that low heart rate (HR) may also relate to use of certain medications (e.g. beta blockers) in diseased individuals. Thus in the discussion section the authors should discuss the changes observed in ECG in HR and HRV between the heathy volunteers dataset and the valvular diseases dataset in the light of these four aspects. For instance, the healthy volunteers are much younger than valvular disease group and this may reflect to HR and HRV and on the other hand patients in valvular disease dataset may have medications which might affect to HR/HRV and also different gender distribution in the two datasets may affect to average HR/HRVs etc. I would recommend to add a few additional sentences about these issues to the discussion section (potentially supplemented with a couple of new citations).

Answer: We agree that the HRV decreases with age and some cardiovascular health conditions.
The data set from patients with VHD contains data collected before the treatment.
We did not assess the use of medication because no such information was available in each data set. We have included the age, gender and health status differences between these two groups
and the use of other measurement devices.

-As a last section of the manuscript (section 5) I would recommend to add conclusions briefly summarising the main findings of this paper.

Answer: The conclusions section has been added to summarize the paper.

Typo corrections:
-In Figure 4 Tachogramm word there is a typo (extra "m").

Answer: corrected

-Line 192 (Discussion): ...SDNN, (ECG and SCG) pNN50 (ECG), VLF (GCG)... There is probably missing comma between some of the words.

Answer: corrected

-In section 2.3 lines 139-140 there is a text: "The very low-frequency band of the HRV spectrum was defined as 0.0033–0.04 Hz, low frequency band was defined as 0.04–0.15 Hz, and the high-frequency band was defined as
0.15–0.4 Hz [54,68].". This sentence should be revised for clarity.

Answer: This sentence was paraphrased as "The frequency bands of the HRV spectrum were defined as follows:
very low-frequency band was defined as 0.0033--0.04~Hz,
low frequency band was defined as 0.04--0.15~Hz,
and the high-frequency band was defined as 0.15--0.4~Hz [54,68]."

-The paper should be carefully read through once more in order to correct any other possible typos. 

Answer: typos corrected

Reviewer 2 Report

The paper deals with the comparison of the time and frequency domains, and nonlinear HRV indices applied to the electrocardiograms, seismocardiograms (SCG signals), and gyrocardiograms (GCG signals). The approaches are tested in healthy and volunteers valvular heart diseases. The idea of the paper is interesting but the paper lacks novelties. Also, it is hard to follow the paper and draw any kind of conclusion. The specific comments follow.

1.       The paper has the form of a review paper but with a humble number of approaches described. It is not clear why those specific approaches are observed for the analysis of the considered signals. There are a number of approaches used for biomedical signal processing and a more extensive description should be provided. Also, it is hard to follow the usability of the presented approaches for the concrete type of signal.

2.       The number of figures is large, providing little information about the presented approaches. The choice of what will be presented in the figures should be done more carefully and systematically.

3.       The tables with some of the results are provided in Section 2, before the Results section. It is not clear why the results are separated into two sections. In my opinion, the Results sections should be left for the concrete results and in the sections Materials and Methods observed approaches should be described.

4.       There is no theoretical background behind the mentioned approaches. Each approach and method which is used in the paper should be mathematically described.

5.       It should be visually better if the length of the signal corresponds to the length of the x-axis. 

Author Response

Thank you for considering the review of our manuscript. Our responses to your comments are as follows:

1. The main objective of the paper was not reviewing the available approaches to heart rate variability analysis in cardiac mechanical signals. If so, the objective and structure of the paper would suggest the form of a review paper.

2. The role of the figures is an illustration of the analyzed data.

3. These tables were moved to an appendix and the overall demographic data of the subjects from both datasets were placed in a separate table.

4. In this study we have applied well-known methods of heart rate variability analysis that are thoroughly described in appropriate references (cf. the recommendations of HRV analysis, Pan-Tompkins algorithm, heartbeat detection in cardiac mechanical signals based on locations of QRS complexes in ECG). The only exception is the Lomb periodogram that was used in frequency domain heart rate variability analysis.

5. The length of the signal has been trimmed to the length of the x-axis.

Reviewer 3 Report

Heart rate variability can be a sign of current health issues or upcoming problems. But it is influenced by different factors and is still difficult to interpret. If electrocardiograms maintain a significant role in the evaluation of subjects with cardiovascular issues, seismocardiograms and gyrocardiograms do not add any valuable data in a clinical setting. 

Author Response

I agree that heart rate variability may indicate current health issues or upcoming problems,
is influenced by many different factors, and remains difficult in interpretation.
Based on the rising numbers of published studies on clinical applications of seismocardiography
and gyrocardiography, providing the information on hemodynamics, heart contractility, condition of heart valves and the operation of at least two companies that offer the diagnosis of cardiovascular conditions based on the two aforementioned techniques, I must disagree with the opinion that seismocardiograms and gyrocardiograms do not add any valuable data in a clinical setting.

Round 2

Reviewer 2 Report

The authors responded to my comments. I suggest accepting the paper for publication.